# Molecular Diversity of *Mycobacterium avium* subsp. *paratuberculosis* in Four Dairy Goat Herds from Thuringia (Germany)

**DOI:** 10.3390/ani13223542

**Published:** 2023-11-16

**Authors:** Chris Pickrodt, Heike Köhler, Udo Moog, Elisabeth M. Liebler-Tenorio, Petra Möbius

**Affiliations:** 1Institute of Molecular Pathogenesis, Friedrich-Loeffler-Institut, Federal Research Institute for Animal Health, Naumburger Straße 96a, 07743 Jena, Germany; chris.pickrodt@fli.de (C.P.); elisabeth.liebler-tenorio@fli.de (E.M.L.-T.); 2Thuringian Animal Diseases Fund, Sheep and Goat Health Service, Victor-Goerttler-Straße 4, 07745 Jena, Germany; umoog@thtsk.de

**Keywords:** MAP, SNP-based assay, MIRU–VNTR, SSR analysis, MAP-C genotypes, fecal, tissue, environmental samples, transmission, small ruminants

## Abstract

**Simple Summary:**

Paratuberculosis (Johne’s disease) is a chronic granulomatous enteritis that affects domestic and wild ruminants worldwide, mainly dairy and beef cattle, but also small ruminants such as goats and sheep. To gain a better understanding of the disease epidemiology and to develop effective control strategies, it is useful to determine the strain diversity of the causative agent, *Mycobacterium avium* subsp. *paratuberculosis* (MAP). The aim of this study was to determine the diversity of MAP isolated from four goat herds affected by paratuberculosis in Thuringia (Germany), as well as the detailed distribution of genotypes among the animals and their environment in one herd (herd 1). A combination of three methods was used to genotype isolates from fecal samples of infected animals, from various intestinal and other tissues of clinically affected animals, and from environmental samples. The six MAP-C genotypes identified could be assigned to five different phylogenetic subgroups. The results suggest individual infection strains within each herd. In herd 1, one predominant strain was found, and two strains occurred sporadically. The genotypes detected were not goat specific. Introduction of MAP into the herd through uncontrolled animal trade or environmental exposure to MAP in barns that were previously used by cattle with unknown paratuberculosis status are suspected as possible sources of infection.

**Abstract:**

This study investigated the intra- and inter-herd diversity of *Mycobacterium avium* subsp. *paratuberculosis* (MAP) isolates from four goat herds in Thuringia (Germany) that were affected by paratuberculosis for several years. The main focus was on the characterization and distribution of genotypes among animals and the environment of goat herd 1. This study included 196 isolates from the feces of 121 infected goats, various tissues from 13 clinically diseased goats, 29 environmental samples from herd 1, and additionally, 22 isolates of different origin from herds 2 to 4. The isolates, sampled between 2018 and 2022, were genotyped using short-sequence-repeat (SSR) analysis, mycobacterial-interspersed repetitive units–variable-number tandem repeat (MIRU–VNTR) analysis, and a single nucleotide polymorphism (SNP)-based assay for phylogenetic grouping. All the isolates belonged to the MAP-C group. In herd 1, one predominant genotype was determined, while two other genotypes were identified very rarely and only in fecal and environmental samples. One of three further genotypes was found in each of herds 2 to 4. The assignment of genotypes to different phylogenetic clades suggested six different infection strains. The results indicated no epidemiological links between the examined herds. Based on the current MAP genotyping data from Germany, possible sources of infection are MAP-contaminated barns previously used by infected cattle and the purchase of sub-clinically infected goats.

## 1. Introduction

*Mycobacterium avium* subsp. *paratuberculosis* (MAP) is the causative agent of paratuberculosis (Ptb) or Johne’s Disease, an incurable chronic enteritis of domestic and wild ruminants as well as camelids and some other non-ruminant species with less clear symptoms [1,2,3,4]. The disease is characterized by therapy-resistant diarrhea, progressive emaciation, and a decrease in milk production, finally leading to death [5]. The clinical symptoms appear after an incubation period of up to several years in the final stage of the disease [6].

In small ruminants, softening of feces is observed in individual animals, but severe watery diarrhea, typical for cattle, occurs very rarely [1]. There is evidence that goats are more susceptible to MAP than sheep [7] and have a shorter clinically unapparent interval, faster disease progression compared to sheep and cattle [7,8], and often show bacterial shedding before the peripheral immune response is measurable [9].

During the subclinical stage of Ptb, the pathogen may be shed intermittently in low to high concentrations in the feces [10], thus contaminating the immediate environment of animals in the barn [11,12] and on pasture [13], or at feeding sites of wild animals [14]. MAP has a high tenacity in manure, soil, grass, and water [15,16]. Animals become infected mainly in the first weeks of their life via the fecal–oral route upon ingestion of contaminated milk and other feed, water, or oral contact with contaminated surfaces [5,17,18]. However, adult animals can also be infected with MAP, as described for cattle and sheep [18,19,20,21]. There is evidence that some individual animals are inherently susceptible to MAP infection, regardless of age [21].

In Germany, in addition to many cattle herds [22], goat herds are also affected by paratuberculosis [23,24,25], as has been described for other countries worldwide [3,26]. Over the last few decades, the global dairy goat population has steadily increased, as has global goat milk production, with regional variations [27]. Goats are traditionally managed differently than cattle, and they graze in expansive enclosures or not enclosed at all [28]. The species (goat) and the absence of perimeter fencing for livestock are considered the main risk for MAP exposure in specific regions [29]. In countries with increased goat milk production, dairy goat farming systems are still mainly managed semi-extensively or semi-intensively, but these conditions provide ample opportunities for the transmission of MAP [1]. Several management practices (poor kidding/birthing hygiene, administration of bulk colostrum or milk, deficits in rearing young animals, inappropriate ventilation) and co-infections with other pathogens have been proposed to facilitate the spread of MAP or MAP infection [26,30,31].

Numerous studies have characterized MAP isolates from different host species and regions of the world to reveal possible transmission routes, persistence of infection, the diversity and evolution of MAP, and to find differences in pathogenicity. A variety of single or combined molecular genotyping methods have been used for this purpose, summarized in [32,33,34], including analyses based on whole genome sequencing (WGS), which provides the highest resolution and can reliably identify phylogenetic relationships [35,36,37]. Depending on the applied method, previously described or novel distinct genotypes have been identified, or isolates have been assigned to defined or novel phylogenetic groups and subgroups by comparing WGS-based single nucleotide polymorphisms (SNPs) [37,38]. Recently, Fawzy et al. [39] recommended a hierarchical genotyping approach including a combination of mycobacterial interspersed repetitive unit variable number tandem repeat (MIRU-VNTR) analysis [40] and an SNP-based assay [38], resulting in fine discrimination of MAP from animals within limited geographical regions.

MAP genomes have been divided into two main groups with different designations in the literature: C-type/Type II (MAP-C, cattle type), with the main subgroups Type C and Type B (bison type); and S-type (MAP-S, sheep type), with the main subgroups Type I and III [33,36,38]. Goats are susceptible to both C-type and S-type strains [1,36,41,42]. In previous studies, goat isolates originating from different herds in Spain or from individual goats in some European countries showed a high diversity [43,44,45] and belonged to various phylogenetic subgroups, together with cattle or sheep isolates [36]. However, most previous studies investigating the intra-herd or inter-herd diversity of MAP or discussing the possible transmission routes of the disease have focused on cattle, sheep, or deer, but not on goats [39,46,47,48,49,50,51,52,53]. Currently, no genotyping data of goat isolates are available from Germany.

In this study, various MAP isolates from goats in Thuringia (Germany) were characterized using the hierarchical genotyping method recommended above, as well as short sequence repeat (SSR) analysis. The aims of this study were: (1) to investigate the intra-herd diversity of MAP in one goat herd, including the immediate environment in the housing, and to determine which strains caused clinical disease in some of the goats; (2) to reveal inter-herd diversity in paratuberculosis-affected dairy goat herds in Thuriingia and provide evidence for epidemiological associations between some of these herds; and (3) to compare the genotypes identified, including their assignment to phylogenetic groups with genotypes previously determined in other hosts or regions.

## 2. Materials and Methods

### 2.1. Study Herds and Animals

Four dairy goat herds, situated in different counties of Thuringia, a federal state of Germany (Figure 1), were included in this study. Herds 1 and 2 kept goats of the breed “Thüringer Wald Ziege”, while herds 3 and 4 mainly consisted of the breed “Saanenziege” (Saanen goat). These herds were monitored by the Sheep and Goat Health Service of the Thuringian Animal Diseases Fund.

Most of the isolates analyzed in this study originated from herd 1. This herd was an organic dairy herd with approximately 400 lactating goats. Herd 1 was established as early as 2002, with a mix of goats, sheep, and cattle kept in a former cattle barn from a herd with an unknown Ptb status (Barn A). Later, the focus changed to solely dairy goat husbandry with their own reproduction. Over the years, several goats from different breeders with unknown Ptb statuses were introduced into the herd to increase the herd size and genetic pool. The last time adult goats (female, n = 35) were purchased from a herd with an unknown Ptb status was in 2011. The purchased goat bucks always originated from Ptb non-suspect herds. Between 2014 and 2015, all the goats and kids were moved to another old cattle barn (Barn B) with an unknown Ptb history before the herd was moved to its current housing, which was also a former dairy cattle barn (Barn C), in 2016. During the data acquisition for this study, it was subsequently revealed that in an initially anonymous screening study in 2011, four out of five of the environmental samples from Barn C had been tested positive for MAP based on their cultures. The corresponding cattle herd was removed in 2012, but eight of the dairy cattle remained on the pasture of the farm until 2016. Barn C was not disinfected after destocking, and the liquid manure remained under the slatted floor in the manure channel. The goat bucks and kids from herd 1 were introduced to this environment in June 2016. In August 2016, the cattle manure was removed, the manure channel was cleaned, and the barn was disinfected, but not using a mycobacterial disinfectant. In early December 2016, the lactating goats were moved into Barn C. Beginning in August 2016, herd 1 had problems with pseudotuberculosis. The last purchase of kids (*n* = 28) was in 2021 from a Ptb non-suspect dairy goat herd. 

MAP was first detected culturally in herd 1 in January 2018 in the tissues of seven clinically affected goats after necropsy. A paratuberculosis control program was implemented shortly after this diagnosis, including vaccination of the entire herd using an inactivated vaccine (Gudair, CZ Vaccines, O Porriño, Spain). In February 2018, MAP was found in 89 of 333 fecal samples from adult female goats based on their cultures, as described further below. In addition to the isolates obtained from the sampling in 2018, fecal isolates recovered from biannual herd samplings in 2020, 2021, and 2022 were also included in this study. Furthermore, the clinically diseased goats from herd 1 were sampled after necropsy in 2018, 2021, and 2022, and MAP was isolated from different tissues. The age of the goats sampled ranged from <1 to 11 years. Most of the goats were two to four years old when MAP was successfully isolated and genotyped. The MAP isolates originated from the goats that were born between 2014 and 2020, with 2014 and 2015 as the focus. Furthermore, environmental samples from various locations in Barn C were sampled in 2020, 2021, and 2022 and cultivated, and if MAP isolation was successful, these isolates were characterized the same way as the isolates from the feces and tissues. 

Herd 2 was established in 2016 through purchases from herds of three different breeders. This included 35 female kids (12 weeks old) from herd 1 with an unknown Ptb status at that time, which were born in February and March 2016 and exclusively housed in Barn B. The purchased adult goat bucks originated from two Ptb non-suspect herds. The goats were kept in a newly built barn. However, in previous years, cattle with unknown MAP statuses grazed on some pastures of this farm. MAP was first detected in November 2018 using the fecal culture of samples from several clinically non-suspect goats. Fecal samples from regular whole herd samplings in 2019 and 2022 were examined in this study.

Herd 3 was established in the early 1990s, first in a former sheep barn and then in a renovated dairy cattle barn. The young goats originated from different small goat and sheep herds with unknown Ptb statuses from Thuringia in Germany. In addition, approximately 600 goats were purchased from different breeders in Switzerland (1997) and in France (2014) with unknown Ptb statuses. In 2004, MAP was culturally confirmed for the first time in tissues from a clinically diseased female goat after necropsy.

Herd 4 was established in 1998. The herd was kept in two parts on separate locations. The goats originated from different breeders in The Netherlands, both without consideration of their Ptb status. MAP was first detected in May 2017 by fecal culture of samples from several clinically suspect goats.

MAP isolates from fecal samples collected in 2017 from herds 3 and 4 as part of another research project were included in the study. Furthermore, bedding samples collected in 2017 and fecal bulk samples collected in 2021 and 2022 for monitoring purposes were analyzed. In addition, isolates from tissues originating from a clinically affected goat of herd 4 necropsied in 2020 were examined.

### 2.2. Sample Collection and Cultivation of MAP

#### 2.2.1. Fecal Samples

Fecal samples from herd 1 were collected at nine sampling events between 2018 and 2022. Each sample was collected from the rectum of the goat using a fresh glove, transferred to the laboratory, and stored at 5 °C when processed within the next days or at −20 °C if this was not feasible. Fecal samples from herds 2 to 4 were collected between 2017 and 2022 in the same way.

Bacterial culture was performed on Herrold’s Egg Yolk Agar (HEYM, Becton Dickinson, Sparks, MD, USA) supplemented with Mycobactin J and ANV (Amphotericin, Nalidixic acid and Vancomycin) for all fecal samples according to the official diagnostic manual of diagnostic procedures published by the Friedrich-Loeffler-Institut [54], or as described previously [55], but with some modifications. A total of 3 g of feces was decontaminated over 48 h at room temperature (RT) using 30 mL of 0.75% hexadecyl pyridinium chloride (HPC) solution (Sigma Aldrich, Taufkirchen, Germany). After discarding the supernatants, 200 µL of the remainder was transferred to each of the three slopes of the growing medium. The slopes were incubated at 37 °C for up to six months and examined every second week. Visible MAP colonies appeared after 4 to 6 weeks at the earliest. The presence of MAP in characteristic colonies was confirmed using subspecies-specific PCR after DNA isolation from the colony material, as described in Section 2.3.

The culture-positive MAP isolates from the feces used in this study of herd 1 (*n* = 133) are listed in Appendix A, and of herd 2 (*n* = 7), herd 3 (*n* = 3), and herd 4 (*n* = 4) in Appendix A. At least one isolate from each animal in herd 1 with successful MAP cultivation during the study period was genotyped.

#### 2.2.2. Tissue Samples after Necropsy

Almost all of the tissue samples originated from animals from herd 1, with the exception of two tissue samples from a goat from herd 4. Thirteen female goats were removed from herd 1 due to clinical symptoms of Ptb, e.g., severe progressive weight loss, intermittent diarrhea, and shedding of MAP. They were euthanized for scientific purposes according to §4, sentence 3 of the German Animal Welfare Act, and they were necropsied for macroscopic examination and sampling for histology and microbiology.

Tissue samples were collected from four representative sites of the jejunum and from the ileum, colon, mesenteric lymph nodes (Lymphonodi [LNN] mesenteriales, LNN ileocolici), colonic lymph nodes (LNN colici), liver, hepatic lymph nodes (LNN hepatici), mammary gland, supramammary lymph nodes (LNN inguinales superficiales), and other parenchyma (spleen, kidney, lung, heart). In all of these goats, lesions were found in the small intestine and mesenteric lymph nodes. Macroscopic inspection revealed thickened intestinal walls and nodular mucosa affecting at least two-thirds of the jejunum. This was confirmed by histology as moderate to severe granulomatous enteritis with few to no acid-fast bacteria (AFB) in five of the goats (paucibacillary) and many AFB in eight of the goats (multibacillary). Multiple calcified granulomas and/or granulomatous infiltrates were present in the mesenteric lymph nodes of all the goats.

The fat and connecting tissue were removed from the lymph nodes and other tissue. The intestine was opened and the ingesta was removed. One gram of sample was cut from different locations, minced with scissors, and transferred into a plastic bag containing 7 mL of 0.9% HPC solution. The samples were homogenized in a stomacher for 6 min, transferred to a 50 mL tube, and agitated on a shaker at 200 rpm for 10 min at RT. Afterwards, they were incubated in an upright position for 24 h at RT in the dark. After centrifugation at 1880× *g* for 20 min at RT, the supernatants were discarded, and the pellet was re-suspended in 1 mL of sterile phosphate-buffered saline (pH 7.2). A total of 200 μL of the pellet was transferred on each of the four slopes of HEYM. The cultures were further treated as described above.

The MAP isolates used for genotyping originated from the jejunum, ileum, mammary gland, spleen, and different lymph nodes (LNN mesenteriales, LNN ileocolici, LNN colici, LNN hepatici, LNN inguinales superficiales). All of the isolates were from different tissues. Their host origin and the year of isolation are presented in Appendix A.

#### 2.2.3. Environmental Samples

Environmental samples (*n* = 256) were collected in the goat barn of herd 1 at eight different timepoints between 2020 and 2022 within the context of another study [12]. Cultivation of bedding, dust, and roughage from the feeding place and trough water was performed as described in detail elsewhere [12]. Briefly, 50 g of bedding and feed samples were soaked in distilled water overnight. The fluid was centrifugated and the supernatants were discarded. The dust was bloated with distilled water overnight and subsequently filtrated. The water was the centrifugated to obtain the sediment. The sediment pellets of all the sample materials were further processed the same way as described for the fecal samples, with the modification that for the decontamination of the dust and water samples, only 5 mL of HPC was used. Cultivation was performed as described above for the fecal samples.

The MAP isolates from all of the culture-positive environmental samples (*n* = 29) from the goat barn (Barn C) of herd 1 are listed, together with their respective collection sites, in Appendix A. These isolates, as well as six isolates originating from the bedding materials of different localizations in the goat barn of herd 4 (see Appendix A), were included in this study.

### 2.3. DNA Isolation and MAP Identification

Mycobacterial DNA was obtained using the method previously described in reference [56]. Extraction started with suspending a loopful of bacterial colony material in 100 µL of distilled water, followed by heating for 20 min at 80 °C, ultrasonication (35 Hz) for 10 min, another heating step for 10 min at 100 °C, and then centrifugation for 5 min at 12,000 rpm (15,300× *g*). The supernatant was transferred to another tube and was centrifuged again for 5 min at 12,000 rpm. The concentration of the DNA in the supernatant was determined using a NanoDrop 1000 spectrophotometer (Thermo Fisher Scientific, Wilmington, DE, USA).

MAP was confirmed by the detection of the subspecies-specific insertion element IS900 according to Englund et al. [57]. Mixed isolates with other *Mycobacterium avium* subspecies were excluded by a lack of IS1245 using PCR, described by Guerrero et al. [58]. Furthermore, the subspecies-specific genotyping profiles determined from the MIRU-VNTR analysis verified the subspecies and the MAP type identification (see Section 2.4.1).

### 2.4. Genotyping of MAP

A total of 133 fecal MAP isolates were randomly chosen for genotyping from 121 out of the 186 goats in herd 1 that were detected to be MAP-positive based on their cultures between 2018 and 2022. The MAP isolates from 13 of the goats with clinical Ptb symptoms selected for genotyping originated from one to four different tissues per goat and included those from the small intestine, various lymph nodes, and occasionally from tissues of other organs. All of the isolates, as well as those from all the environmental samples of herd 1 and from all the samples of the other herds, were genotyped using three techniques: MIRU-VNTR analysis, an SNP-based assay, and SSR analysis. For these analyses, the DNA extracts isolated as described in Section 2.3 were used.

#### 2.4.1. MIRU-VNTR Typing

Using this method, differences in the number of tandem repeat sequences were detected using single PCR reactions targeting eight defined mini-satellite loci in the MAP genomes: MIRU-VNTR Loci 292, X3, 25, 47, 3, 7, 10, and 32 [40,59]. Different repeat numbers describe different alleles at these specific loci. These numbers were arranged according to the mentioned loci order, and based on the INMV classification database (http://mac-inmv.tours.inra.fr/ (accessed on 14 June 2023)), the so-called INMV types were determined. This method can be used for isolates belonging to different subspecies of *Mycobacterium avium*, which show subspecies-specific INMV types [60].

The same primers and PCR conditions described by Thibault et al. [40] were used with minor modifications according to Möbius et al. [61] as follows: approximately 50 ng of DNA was amplified in a 20 µL reaction volume containing 2 µM of each primer, 1 × reaction buffer containing 1.5 mM MgCl_2_, 1 × Q-Solution (Qiagen, Hilden, Germany), 100 µM of each deoxynucleotide triphosphate (Qiagen, Hilden, Germany), and 0.5 U of HotStarTaq polymerase (Qiagen, Hilden, Germany). The PCR started with an initial denaturation step at 94 °C for 15 min, followed by 35 cycles of 96 °C for 30 s, 58 °C (MIRU 292, VNTR 25) for 1 min, and 72 °C for 30 s, with a final extension step at 72 °C for 10 min. The annealing temperature for MIRU X3 was 62 °C, and for VNTR 10, it was 54 °C. The annealing temperatures of the other target loci were applied according to [40]. Based on the amplicon sizes of the amplified target sequences analyzed using a 1.5% agarose gel, the number of repeats were determined using an allele-calling table published by Radomski et al. [59]. Amplicons presenting as single bands show that there was no mixed genotype.

Additionally, to identify polymorphisms at VNTR locus 7 for isolates with unusual amplicon sizes, the PCR products were purified using the QIAquick PCR purification kit (Qiagen, Hilden, Germany) according to the manufacturer’s instructions and sequenced by an external sequencing service. The resulting sequences were compared with a reference sequence, as described in reference [62].

#### 2.4.2. SNP-Based Assay

This PCR-based method with subsequent restriction enzyme digestion or sequencing of the amplified product allows for the assignment of MAP isolates to 14 phylogenetic groups within the MAP population, each defined by a specific SNP [38]. These SNPs are mutually exclusive, i.e., if an isolate possess one particular SNP, it lacks all the other ones [38]. As recommended in [39] (see Appendix A), a MIRU-VNTR analysis was performed first. Based on the detected INMV types, specific PCRs were prioritized for the SNP-based analysis to reduce the number of SNP-PCRs for each sample. According to this work, the SNPs were analyzed stepwise, and analysis was stopped once a phylogenetic group-defining SNP was identified.

For single-plex PCRs, the primers as published were applied, and the SNP-PCRs were named based on their primer designation according to reference [38]. A minimum of 50 ng of DNA was amplified in a 50 µL reaction volume containing 2 µM of each primer, 1 × reaction buffer containing 1.5 mM MgCl_2_, 1 × Q-Solution (Qiagen), 100 µM of each deoxynucleotide triphosphate (Qiagen), and 0.5 U of HotStarTaq polymerase (Qiagen). The PCR started with an initial denaturation step at 94 °C for 15 min, followed by 35 cycles of 96 °C for 30 s, 64 °C for 1 min, and 72 °C for 1 min, with a final extension step at 72 °C for 10 min. The original method was modified, with no restriction endonuclease digestion applied, and all the PCR products were sequenced. Approximately 40 µL of the amplified product was purified using a QIAquick PCR Purification Kit according to the manufacturer’s instructions. Sequencing was carried out by Eurofins Genomics GmbH (Ebersberg, Germany). The presence or absence of a specific SNP was analyzed by comparing the sequence result with the genome sequence of MAP K10 (GenBank accession number NC_002944.2) at the defined position according to reference [38] using Geneious Prime 2021.0.1. (https://www.geneious.com (accessed on 11 May 2023)). The original sequence chromatograms were checked for the absence of mixed peaks to exclude mixed genotypes.

#### 2.4.3. SSR Typing

The SSR sequencing method was applied according to [63]. In deviation from the original method, only micro-satellite loci SSR1 (g repeats), SSR8 (ggt repeats), and SSR9 (tgc repeats) were chosen because these loci are among the most informative SSR loci [60,63,64] and the results are stable and reliable [63]. As a result, the number of specific repeats at these three loci was given as a numerical code. The primers were used as published [63]. The PCR mixture and the cycler program were slightly modified as follows: The 50 µL PCR reaction mixture for SSR8 and SSR9 comprised 0.6 µM of each primer, 1 x reaction buffer containing 1.5 mM MgCl_2_, 5% dimethyl sulfoxide, 100 µM of each deoxynucleotide triphosphate, 1 U of HotStarTaq polymerase, and 50 ng of DNA. For SSR1, 2.5 U of Herculase II Fusion DNA polymerase (Agilent Technologies Deutschland GmbH, Waldbronn, Germany) was used to minimize the risk of polymerase slippage at the poly(G) motif, and no dimethyl sulfoxide was used. The PCR for SSR8 and SSR9 started with an initial denaturation step at 94 °C for 15 min, followed by 35 cycles of 96 °C for 15 s, 60 °C for 1 min, and 72 °C for 30 s, with a final extension step at 72 °C for 3 min. For SSR1, the PCR started with 98 °C for 2 min, followed by 30 cycles of 98 °C for 20 s, 60 °C for 20 s, and 72 °C for 30 s, with a final extension step at 72 °C for 3 min. The PCR products were purified using a QIAquick PCR Purification Kit according to the manufacturer’s instructions, and sequencing was performed by an external sequencing company. The resulting sequences were analyzed in silico, and the original sequence chromatograms were checked for mixed peaks to exclude mixed genotypes.

#### 2.4.4. Calculation of the Discriminatory Power of Typing

As a numerical index with the discriminatory power of a combination of MIRU-VNTR, an SNP-based assay, and SSR typing, the Simpson’s Index of Diversity (SID) was calculated, as described by Hunter and Gaston [65]. Only epidemiologically unrelated strains were included in the calculation, i.e., only isolates with different genotypes from the same herd.

## 3. Results

### 3.1. Determined Combined MAP Genotypes

Using MIRU-VNTR typing, five different INMV types were determined for the isolates included in this study: INMV1, 2, 3, 16*, and 33. The amplicons always showed clear individual bands for each MIRU-VNTR locus in the agarose gels, indicating that none of the isolates were mixed isolate with different genotypes. All the isolates were assigned to the C-type group based on the identified MAP-specific INMV types. In addition, none of the isolates belonged to any other subspecies of *Mycobacterium avium*. Furthermore, SNP-PCRs were prioritized, as described in Appendix B. The specific SNPs identified within the MAP genome using the SNP-based assay were used to assign the isolates to the five different phylogenetic groups: Clades 1, 3, 4, and 9 in Subgroup A and Subgroup B (SGB). Using SSR analysis, two different profiles were found: 7g-4ggt-4tgc (7-4-4) and 7g-4ggt-5tgc (7-4-5) at SSR loci 1, 8, and 9. The combination of the individual methods resulted in six combined genotypes for the tested isolates, designated as T1 to T6 (see Table 1).

Genotype T6 included a MIRU-VNTR profile with an unusual sequence at MIRU-VNTR locus 7. The PCR amplicon size at the VNTR 7 locus of several isolates in the agarose gel was slightly below 269 bp, corresponding to five repeats according to Appendix A in reference [59]. Sequencing and in silico analysis revealed three perfect copies of the 22 bp repeat (CGAAATATTCGCCGTGAGAACA) at VNTR 7, but an additional sequence region of 34 bp (CGTGCGGCGAAGGCTGGGCCGGCCCGAAAAGCCA) was identified between repeats 2 and 3. The resulting amplicon size of 259 bp determined through sequencing in the present study was visually almost identical to the amplicon size for the isolates with five repeats. The assignment of isolates with such an unusual sequence at VNTR locus 7 to a specific INMV type using the database (http://mac-inmv.tours.inra.fr (accessed on 14 June 2023)) is not yet clear. Therefore, this result was highlighted with an asterisk (5*), and the respective INMV16 type (32332528) was designated as INMV16* (323325*28).

### 3.2. MAP Strain Diversity in Herd 1

In herd 1, three combined genotypes were identified with different frequencies in isolates from 121 different goats (Table 2). In total, 56 out of 57 fecal MAP isolates of individual animals from 2018 harbored the combined genotype T1 (INMV1/7-4-4/Clade 9). The remaining isolate from 2018 showed genotype T2 (INMV33/7-4-4/Subgroup B), which was isolated again from another goat in 2020 and 2021. Furthermore, genotype T1 was recovered in 36 of the 39 fecal isolates in 2020 (including one isolate from a buck), in 27 of the 29 fecal isolates in 2021, and in 8 of the 8 fecal isolates in 2022 (Appendix A). Fecal isolates from twelve goats originating from two different samplings were genotyped with the same T1 result. In addition, genotype T1 was determined in all the tissue isolates (*n* = 34) from 13 clinically diseased goats necropsied and sampled in 2018, 2020, and 2022 (Appendix A) and in 28 of the 29 environmental MAP isolates (Figure 2, Appendix A). The third genotype, T3 (INMV2/7-4-4/Clade 1), was isolated in 2020 from the feces of one goat (4 years old), an environmental sample from the barn entrance in 2020, and from the feces of another goat (one year old) in 2021 (Appendix A, Figure 2).

### 3.3. MAP Genotypes in Other Goat Herds in Thuringia

The other three Thuringian goat herds investigated harbored three other genotypes (Table 2). In herd 2, seven fecal isolates from 2019 and 2022 all showed the same genotype: T4. In herd 3, isolates originating from two fecal samples of one goat in 2017 and one pooled fecal sample in 2022 revealed genotype T5. Furthermore, in herd 4, isolates from the feces, mesenteric lymph nodes, and jejunum of a total of 4 goats in 2017 and 2020, from six environmental samples in 2017 and from a pooled fecal sample in 2021, all showed genotype T6.

### 3.4. Discriminatory Index of the Genotyping Approach Used

In each goat herd, a different genotype was found in addition the two genotypes in herd 1. Based on these six epidemiologically unrelated strains (Table 2), the discriminatory power of typing by using the combination of MIRU-VNTR, SSR analysis, and an SNP-based assay was calculated and resulted in a discriminatory index of 1.

## 4. Discussion

This study provides the first insights into the molecular diversity of MAP isolated from naturally infected goats in one federal state in Germany, including 203 isolates from 147 dairy goats and 35 environmental isolates from 4 herds in the federal state Thuringia. Three genotyping methods were used in combination. The previously published stepwise hierarchical typing workflow, combining MIRU-VNTR typing and an SNP-based assay [39], was successfully applied, and the typing effort of the SNP-based assay was reduced. The main findings of this study are as follows: (1) the genotypes of the isolates were different in the four goat herds tested, indicating that they were not epidemiologically linked. Altogether, six combined MAP-C genotypes were identified. (2) In herd 1, where most of the isolates originated from, one dominant genotype, and sporadically, two additional genotypes, were found. (3) MIRU-VNTR typing resulted in types INMV1, 2, 3, 33, and additionally, a type with a specific structure at locus VNTR 7, designated as INMV16*. No mixed MAP genotypes and no other subspecies of the *Mycobacterium avium* complex were identified. (4) Using SSR analysis, two profiles were detected (7-4-4, 7-4-5). (5) Using the SNP-based assay, the isolates were assigned to five different phylogenetic groups (Clades 1, 3, 4, and 9 in Subgroup A, and Subgroup B). In previous studies, these subgroups and phylogenetic clades included isolates from different hosts, regions, and countries worldwide.

### 4.1. Genotyping Results

The sampled Thuringian goat herds, all infected by C-type (Type II) strains, were kept solitary; not together with sheep or cattle. In Germany, S-type (Type III) strain isolation so far only has been described in a few sheep [66]. The current study focused on C-type strains using solid HEYM medium with Mycobactin J for MAP isolation, which is less sensitive for isolation of S-type strains [67,68]. In studies characterizing MAP isolates or MAP-DNA from goats in Spain, Mexico, Canada, and Switzerland, the detected predominant type was also the C-type; however, S-type strains/genomes (Type I/III) were also identified [41,69,70,71]. The goat herds tested were infected either by C-type or S-type strain(s) only [41], or both types were identified in one herd [70]. In Spain, many farms with native goat breeds were infected with S-type strains, and a larger number of these goats were in contact with sheep that showed no signs of infection [41].

Worldwide, only a few previous studies have been carried out in goats investigating the diversity of MAP at the herd level [43,70,72]. The first common methods for genotyping of MAP isolates were pulsed-field gel electrophoresis (PFGE) and restriction fragment length polymorphism coupled with hybridization to IS900 (IS900-RFLP) analysis. De Juan et al. [43] identified 15 combined genotypes (14 × Type II, 1 × Type III) in 44 caprine isolates from Spain, Scotland, and Norway, including high intra-herd diversity with six genotypes in a single herd from Spain, detected using PFGE in combination with IS900-RFLP. In contrast, only one genotype (Type II) could be found in 25 caprine isolates from one individual herd in Germany using IS900-RFLP [72]. In the meantime, PFGE and IS900-RFLP have been replaced by PCR- and sequence-based methods (MIRU-VNTR typing, SSR-analysis), which are faster and less labor-intensive. In addition to one S-type isolate, Bauman et al. [70] found three genotypes in 21 C-type isolates from 11 dairy goat farms in Canada using a combination of MIRU-VNTR and SSR-typing (analyzing only the ggt SSR locus 8). Almost all of the isolates belonged to the combined subtypes INMV1/4ggt and INMV2/5ggt. This low diversity was attributed to the limited discriminatory power of the typing approach used, which is particularly true for isolates with INMV1 and INMV2 [50]. On the other hand, it could also be due to the fact that there were only two main infection lineages in these Canadian goat herds. More target regions, as in the present study, might have led to a higher discriminatory power of typing and thus to more genotypes. Apart from this, WGS-based data provided the highest resolution for distinguishing MAP as a monomorphic pathogen on a regional and global scale and allowed for the detection of phylogenetic relationships between isolates within the MAP population [50,73,74]. However, a study on Canadian cattle isolates showed that SNPs alone were not sufficient to track MAP transmission due to the slow growth rate of MAP and because the extensive movements of affected animals between different provinces were not well documented [35]. Furthermore, using WGS data, a comparison of genotypes between different studies is complicated, as it would require renewed analysis of the raw data of all included genomes. Alternatively, the recently established PCR-based SNP assay [38] allows strains to be assigned to previously defined known phylogenetic groups and subgroups without the need for WGS of isolates. Combining this assay with MIRU-VNTR typing and the SSR analysis in this study allowed for successful discrimination between isolates from different goat herds in Thuringia (Germany).

MIRU-VNTR types INMV1 und INMV2 are predominant MAP genotypes isolated from cattle and goats in Europe, Canada, the United States, and several countries in Latin America [36,40,50,70,75,76]. Consistent with this, in the present study, INMV1 was found in almost all the samples from herd 1, and INMV2 was found in all the isolates from herd 2, in four isolates from two goats, and in one environmental sample in herd 1. In addition, the goat isolates from Thuringia harbored INMV3, INMV16*, and INMV33. INMV3 and INMV33 are commonly found in Europe and South America [39,60,77,78,79,80,81]. Type INMV16* was a special case, because sequencing of the VNTR 7 locus amplicon did not reveal the expected five repeats (based on amplicon size) but instead an unusual sequence comprising two repeats and an additional 56 bp sequence. This additional sequence was described before as a second repeat sequence, including 34 bp and a sequence identical to the 22 bp VNTR 7 repeat [62]. The respective complete sequence at the VNTR 7 locus was designated as profile E [62]. Possibly, type INMV16, previously found in bovine isolates from France, Argentina, and Germany [39,77,79,82], corresponds to type INMV16* in the present study because the PCR products were similar. Unfortunately, it was not possible to assign this MIRU-VNTR profile with an unusual amplicon size and the underlying sequence differences at specific loci to defined INMV types using the database or previous studies.

Furthermore, the SSR analysis at SSR loci 1, 8, and 9 revealed two out of 13 profiles/alleles (based on these three loci) that were previously identified in MAP isolates originating from different hosts in the United States [63]. The predominant SSR profile (7-4-4) was found in combination with MIRU-VNTR types INMV1, 2, 3, 16*, and 33; the profile 7-4-5 was found with INMV2 only. In one previous study including 12 goat isolates, the profile 7-4-4 was also found to be predominant. It was detected in nine goat isolates from France in combination with INMV1 (*n* = 7) and INMV2 (*n* = 2) and in one goat isolate from the Netherlands in combination with INMV33 [60], as in this study. Two additional goat isolates from France harbored the 7-4-5 profile, identical to our second SSR profile, but in combination with INMV1 and not with INMV2. The combination of the 7-4-4 profile with INMV1, 2, 3, or 33, found in the present study was also detected in several cattle isolates from France, the Netherlands, Argentina, Czech Republic, Slovenia, and Germany, and in combination with INMV1 or 2 in a few deer isolates [46,60]. Regardless of their assignment to phylogenetic clades, these results suggest that the majority of MAP genotypes from the studied goat herds in Thuringia belonged to the MAP population that is common in infected cattle and other ruminants in Central European countries.

Previously, based on WGS–SNP analysis, a set of MAP isolates originating from different hosts and countries worldwide was compared. The isolates were distributed within a phylogenetic tree to the known major S-type (Type I and III) and C-type (Type II) groups and various subgroups [36]. Type II included Subgroup A with 10 clades, Subgroup B (SGB), and Type B (bison type) [36,38]. These defined assignments of the MAP isolates to phylogenetic groups, subgroups, and clades were used for establishing the SNP-based assay based on 14 discriminative SNPs by Leão et al. [38], and this assay was applied in the present study. In contrast to the misidentification of Australian S-type isolates and the incorrect differentiation of C-type isolates into subgroups A or B [83], this method works well for German C-type isolates. Overall, the goat isolates from Thuringia could be assigned to five of these phylogenetic subgroups: Clades 1, 3, 4, and 9 in Subgroup A, and SGB. In the global MAP set [36,38], these subgroups included isolates from goat (Clades 1, 9, and SGB), cattle (all subgroups), sheep (Clades 1, 3, 4, and 9), deer (Clades 1 and 9), and humans (Clades 3 and 4) from different regions worldwide. The present results indicate a high genetic similarity of the goat isolates from Thuringia with those of the global MAP population.

Most isolates of the global set also showed INMV1 and INMV2 distributed to Clades 1, 2, 8, 9, and 10 (INMV1) or Clades 1, 3, 4, 5, and 10 (INMV2) [36,38]. The assignments of isolates with INMV1 or INMV2 to different phylogenetic clades revealed that the MIRU-VNTR types (especially these two most common types, which differ only at VNTR 292) do not reflect the phylogenetic relationships [36,37,50]. In comparison with WGS-based phylogeny, instances of MIRU-VNTR homoplasy have been observed; MIRU-VNTR typing over- and underestimates the genetic diversity of MAP [36]. Recent studies of modern *Mycobacterium tuberculosis* lineages have shown that the homoplasy of MIRU-VNTR repeats has arisen through stochastic processes in the absence of natural selection [84], which has not yet been investigated in this way for MAP. On the other hand, based on the WGS phylogeny of French animal isolates, a strong association between phylogeny and INMV1 and evidence of an association between phylogeny and INMV2 was found, but not for SSR typing results [73]. All these results suggest that the informative value of MIRU-VNTR typing for phylogeny compared to WGS-based SNP analysis varies depending on the phylogenetic clade to which the isolates belong, on the diversity of the regional origins of isolates, on the time period of sample collection and possibly the phylogenetic analysis methods, including different cut-offs (number of SNPs) for closely related strains and the phylogenetic distance of the included isolates.

German goat isolates from Thuringia with INMV1 were assigned to Clade 9, and isolates with INMV2 were assigned to Clades 1 and 4. Furthermore, goat isolates with INMV3 were also assigned to Clade 4 (this study), as previously reported for cattle isolates from Hesse and Thuringia in Germany [39]. In contrast, isolates (from cattle) with INMV3 in the global set were assigned to Clades 1 and 5 [36,38]. Additionally, isolates harboring INMV33, originating from goats (this study) or from cattle in Spain or Germany [36,38,39], were all assigned to SGB. Furthermore, MAP isolates with the rare genotype INMV16* and belonging to Clade 3 (this study) have not been previously described; however, cattle and human isolates with rare and particularly high repeat numbers at the VNTR 7 locus (INMV16, INMV75, INMV127) have also been assigned to Clade 3 [36,38,39]. However, there are other goat isolates of Type II from different countries worldwide belonging to Clade 7, Clade 10, and Type B [36,38]. These results suggest that within the C-type (Type II) group, there is no host association with phylogenetic clades, and until now, there has been no goat-specific combined genotype.

### 4.2. Epidemiological Analysis and MAP Transmission

The results of combined genotyping targeting different genome regions showed that Ptb in the four Thuringian goat herds was caused by different MAP strains and that herd 1 was infected by three strains. In addition to the predominant genotype T1 (INMV1/Clade 9/7-4-4), few fecal and environmental isolates carried genotype T2 (INMV33/SGB/7-4-4) or T3 (INMV2/Clade 1/7-4-4). Various studies have already shown that in certain goat and cattle herds, multiple MAP infections with different genotypes circulate within the herd [43,46,61,73,85]. As mentioned above, herd 1 was identified as being highly affected by Ptb in early 2018 and was included in a Ptb control program. Ptb-infected goats, shedding the pathogen in large numbers or showing clinical signs, were removed from the herd. This led to a reduction in MAP culture-positive fecal samples from 30.0% in 2018 to 5.2% in 10/2020 and to 1.7% in 05/2022 [12]. In the isolates from the various tissues of the clinically affected goats in herd 1 (born between 2014 and 2018), sampled in 2018, 2020, and 2022, only one genotype (T1) was identified. It is assumed that individual animals have been infected with T1 several times.. The pathogen was disseminated throughout the body and was found not only in the different sections of the intestine, but also in the mammary gland, mammary lymph nodes, spleen, and in the hepatic lymph nodes of individual animals. The spread of MAP to tissues outside the gut has already been described for naturally infected cattle and a goat [71,86] and in experimentally infected goats [55]. The other two genotypes in herd 1 (T2 and T3) were not able to spread further or even to displace the predominant strain, although infected animals remain susceptible to co-infection with other MAP genotypes [20]. Based on the WGS-SNP analysis in another study, some MAP strains were found to be more successful in spreading than others; these were designated as dominant strains [20]. However, there is no evidence that individual strains within the MAP-C type group are more virulent than others. SNPs in various virulence-associated genes in individual strains may be associated with differences in virulence [74,87], but this would need to be demonstrated by in vitro and in vivo experiments. The predominance of T1 could be due to the continuous ingestion of very large amounts of this strain by kids, juvenile, and adult goats over a long period of time, acquired from an environment that is highly contaminated by the previous occupants of the barns and later from high shedders in the goat herd. The high stress induced by the move of the goats to Barn C and the pre-existing problems with pseudotuberculosis may have accelerated the infection processes and the development of clinical Ptb in some animals by the end of 2017. In addition to the detection of MAP in two- to five-year-old goats, genotype T1 was also detected in fecal samples from two seven-months-old goats in 10/2020, born in herd 1. This was possibly caused by the ingestion of MAP during the first hours of life in the highly contaminated kidding area [12]. Furthermore, not only infection of kids but also adult-to-adult contact was assumed to be an important route of transmission, as previously described for dairy cattle [20]. This option is supported by the repeated detection of MAP, particularly at sites with high animal traffic, as previously published [12].

Genotype T2 was only found in the fecal samples of two adult goats, once in 2018 from goat 84,924 (born and purchased in 2011, left the herd March 2018), and twice in 2020 and 2021, from goat 81,024 (born in 2014 in herd 1, left the herd July 2021). It is suspected that either only these two goats were infected with T2 or not all of the T2-infected goats could be identified due to the lack of MAP shedding by 2022. Goat 84,924 was purchased as a kid in 2011 together with other goats with unknown Ptb statuses from a breeder. Some of these other goats, as well as goats born in 2009 and later in herd 1, were found to be infected with genotype T1 in 2018, suggesting different infection events and sources of T1 and T2. Strains with a similar genotype to T2 (but with an unknown SSR profile) have also been previously found in bovine isolates from Thuringia, Hesse, and Saxony in Germany [39].

Furthermore, genotype T3 was detected in 2020 and 2021 in the feces of two goats (born in 2014 and 2020 in herd 1), and in 2020, in an environmental sample collected at the barn entrance. This entrance was utilized twice a day during the grazing season for driving the goats to the pasture. It is possible that the soil was contaminated by the feces of the goats. It is likely that T3 was already present in herd 1 in 2018, as the older goat already tested cultural positive for MAP in the first conducted herd sampling at that time. The younger goat may have been infected via environmental contamination in the kidding box, as this goat in 2020 was born three days after the kidding of the older goat, which was infected with T3. The T1 genotype was predominant in herd 1; however, the true origin of the three found MAP strains could not be determined based on the analyses performed in this study and the available data.

Herd 2, which also kept the breed “Thüringer Wald Ziege”, was infected with a different strain than herd 1: genotype T4 (INMV2/Clade 4/7-4-5). The origin of this strain is unknown. It is possible that the goats were infected by environmental contamination with MAP from cattle on the pastures of herd 2. Herds 3 and 4 kept a different breed of goat (“Saanenziege”, Saanen goats) than herds 1 and 2 and were infected with T5 (INMV3/Clade 4/7-4-4) in herd 3 and T6 (INMV16*/Clade 3/7-4-4) in herd 4. The animals in herd 3 were purchased from several breeders, and the herd had recently used a former dairy cattle barn from a herd with an unknown Ptb status. Only three isolates from herd 3 were tested, but an identical genotype to T5 in 2017 and 2022 suggests that this was the only or the predominant genotype, with an unknown origin. In herd 4, infection strain T6 with the rare MIRU-VNTR profile INMV16* and assignment to clade 3 was identified in all the fecal, tissue, and environmental samples from 2017, 2020, and 2021, suggesting this was the predominant infection strain in this herd. Ptb may have been introduced through the purchase of sub-clinically infected goats.

Furthermore, the presence of different ruminant species with unknown Ptb statuses has been described as another risk factor for MAP infection [26,30]. As described before, herd 1 was established with a mixture of goats, sheep, and cattle in 2002, and herd 3 was first established in a former sheep barn in the early 1990s. This also raises the possibility that there might be an epidemiological link to sheep, in addition to the already discussed one to cattle. Due to the time that has elapsed since the concerned ruminants left the facilities or herds, no further research can be conducted to address this issue. Highly discriminatory genotyping of MAP isolates from a herd of interest, or from different herds with the same origin of kids, combined with appropriate information on the origin of animals, may allow for the origin of an infection to be determined [88]. However, due to the relatively long subclinical phase of Ptb, this is only possible with a time delay. Using WGS-based SNP analysis with a higher resolution for differentiation could identify mixed isolates in individual animals or additional genotypes, as shown in recent studies for individual dairy cattle herds [20,89]. On the other hand, the WGS approach could potentially reveal additional SNPs as a result of evolution in individual herds [90]. However, as the molecular clocks of the specific target regions used for genotyping are different, a combination of methods could still help to identify the different infection lineages and transmission routes of MAP within a limited region and time period.

## 5. Summary and Conclusions

The applied combination of MIRU-VNTR, an SNP-based assay, and SSR-typing was well suited to differentiate MAP isolates from different dairy goat herds in Thuringia (Germany). Each of these goat herds had been infected predominantly clonally with a dominant MAP-C genotype for several years, and herd 1 was additionally infected with two other MAP-C genotypes. These three different strains in herd 1 were most likely introduced at different times. There was no evidence for the transmission of MAP between the goat farms studied. The purchase of sub-clinically infected animals and a MAP-contaminated environment of the goats are suspected as possible sources of the initial MAP introduction in the herds. The exact sources of the MAP introduction and transmission routes could not be determined. This would require knowledge of the Ptb history of all the breeders or farms from which the goats were purchased, the place of birth of the goats, and mother–daughter relationships. Furthermore, isolates from the Ptb-positive herds and from the contaminated environment of any previously used barns or pastures would have had to be genotyped.

The goats were probably not infected with unique or regional MAP strains. The infecting strains were assigned to five different phylogenetic clades, which also included isolates from other countries, different regions, and ruminant species of the global MAP population. The identified combined MIRU-VNTR and SSR profiles have also previously been found in different hosts and regions worldwide. The specific combination of the typing results of the three used methods was compiled for the first time. Further studies could characterize and compare these isolates based on WGS-SNP analyses, reveal the relationship of isolates representing the six identified genotypes, and elucidate the possible evolutionary processes of the individual infection strains within the herds.

## Figures and Tables

**Figure 1 animals-13-03542-f001:**
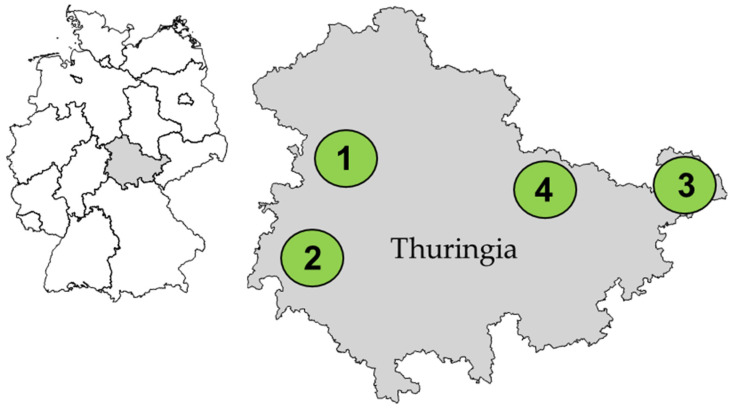
Localizations (1–4) of the examined four goat herds (herd 1 to 4) within the federal state Thuringia in Germany. The maps were generated with © GeoBasis-DE/Bundesamt für Kartographie und Geodäsie (2023).

**Figure 2 animals-13-03542-f002:**
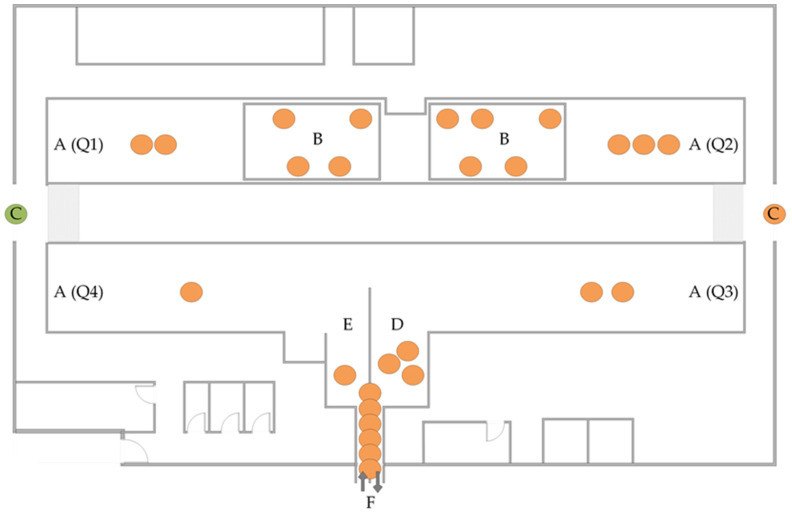
Origins of culture-positive environmental samples within the goat barn of herd 1 (*n* = 29) and determined *Mycobacterium avium* subsp. *paratuberculosis* genotypes. A (Q1–4)—Lactating goat area (stable quadrants 1–4); B—kidding area; C—stable entrance; D—waiting area milking parlor; E—exit milking parlor; F—milking parlor; orange points: genotype T1, green point: genotype T3.

**Table 1 animals-13-03542-t001:** Combined genotypes for *Mycobacterium avium* subsp. *paratuberculosis* isolates from goats and their environment in this study.

Genotype	Phylo-Group ^a^	INMV Type ^b^	INMV Code ^c^	SSR Profile ^d^
T1	9	1	42332228	7-4-4
T2	SGB	33	32522228	7-4-4
T3	1	2	32332228	7-4-4
T4	4	2	32332228	7-4-5
T5	4	3	32332218	7-4-4
T6	3	16*	323325*28	7-4-4

^a^ The phylogenetic groups based on SNP analysis using SNP-based assay according to reference [38] were designated also as Clades (here, Clades 1, 3, 4, and 9) belonging to a larger group named subgroup A; or designated as SGB—subgroup B. ^b^ INMV type according to the INMV database (http://mac-inmv.tours.inra.fr/ (accessed on 14 June 2023)). ^c^ The INMV code comprises the number of tandem repeats (TR) at MIRU-VNTR loci 292, X3, 25, 47, 3, 7, 10, and 32 according to reference [40]. Underlined are repeat numbers different from type INMV2. ^d^ The short sequence repeat (SSR) profile is a combination of the number of SSRs at [G] locus 1, [GGT] locus 8, and [TGC] locus 9 according to reference [63]. * At VNTR 7, in addition to three repeats, an additional sequence region of 34 bp between repeats 2 and 3 was identified. The amplicon size of 259 bp looked like almost five copies; therefore, this result was highlighted with 5*, and the respective code (323325*28) was designated as type INMV16*.

**Table 2 animals-13-03542-t002:** Thuringian goat farms tested for *Mycobacterium avium* subsp. *paratuberculosis* in this study: identified genotypes and origin of isolates (*n* = 220).

Farm	Genotype *	Origin	Isolates (*n*)	Goats (*n*)	Year
1	T1	Feces	127	117 **	2018, 2020, 2021, 2022
		Tissue sample	34	13	2018, 2020, 2022
		Environment	28	-	2020, 2021, 2022
	T2	Feces	3	2	2018, 2020, 2021
	T3	Feces	3	2	2020, 2021
		Environment	1	-	2020
2	T4	Feces	5	5	2019
		Feces	2	2	2022
3	T5	Feces	2	2	2017
		Feces, bulk sample	1		2022
4	T6	Feces	3	3	2017
		Feces, bulk sample	1	-	2021
		Tissue sample	2	1	2020
		Environment	6	-	2017

* Description of combined genotypes T1–T6, see Table 1. ** including one buck. LNN—Lymph node. Underlined—identical genotypes isolated from the same animal in different years (T2) or twice in the same year (T3).

## Data Availability

All data are included in the Appendix A.

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
