# Peer review of "Molecular Diversity of Mycobacterium avium subsp. paratuberculosis in Four Dairy Goat Herds from Thuringia (Germany)"

_animals, 2023, doi:10.3390/ani13223542_

Round 1

Reviewer 1 Report

Comments and Suggestions for Authors

General comments

The manuscript describes the results of MAP genotyping in four goat herds in Thuringia (Germany). It is very well prepared and written, the study designs and methods applied seem to be sound and are described very detailed. Altogether, the manuscript is quit long and technical in some aspects but provides interesting new results. Only very few minor issues were identified during the review which are listed below and should be addressed.

Specific comments

Line 13: The aim of his study was

Line 20-22: introduction of MAP into the herd by uncontrolled animal trade or environmental exposure to MAP in stables that were previously used by cattle with unknown paratuberculosis status are suspected as possible sources of infection.

Line 27: The main focus was

Line 37: The (Our) results indicate…

Line 50: Suggestion: replace “…and death” with “…finally leading to death” or similar

Line 64-65: Please rephrase, what is meant by “animals”? Species? Individuals? Breeds? Genetic lines?

Line 67: Replace “in the world” by “worldwide”

Line 121: Delete “from breeder W.”

Line 136: “MAP was first detected culturally in farm 1 …”

Line 147: At what timepoint the environmental samples have been collected?

Line 115-165: When describing the origin of the purchased goats always add the Ptb status of the herds, if not known please add “unknown Ptb status”. This information is missing for some farms, animals were purchased from.

Line 196: are listed

Line 204: Clinical symptoms

Line 264: How were the samples for genotyping selected?

Line 438: originated from,

Line 449: so far only has

Line 571: The results

Line 514 and 618: Delete “breeder W.”

Line 665: There was no evidence for transmission…

Table 1: Write “Mycobacterium avium subsp. paratuberculosis” in the title, or add a key, as it has to be understandable without connection to the text of the manuscript

Table 2: Include “…tested for Mycobacterium avium subsp. paratuberculosis…” in the title

Author Response

Thank you for reviewing our manuscript. All changes made in the manuscript have been highlighted in yellow.

Reviewer 1)

General comments

The manuscript describes the results of MAP genotyping in four goat herds in Thuringia (Germany). It is very well prepared and written, the study designs and methods applied seem to be sound and are described very detailed. Altogether, the manuscript is quit long and technical in some aspects but provides interesting new results. Only very few minor issues were identified during the review which are listed below and should be addressed.

Specific comments

  • Line 13: The aim of his study was

Was done as recommended.

  • Line 20-22: introduction of MAP into the herd by uncontrolled animal trade or environmental exposure to MAP in stables that were previously used by cattle with unknown paratuberculosis status are suspected as possible sources of infection.

Was done as follows:

“Introduction of MAP into the herd by uncontrolled animal trade or environmental exposure to MAP in barns that were previously used by cattle with unknown paratuberculosis status are suspected as possible sources of infection.”

  • Line 27: The main focus was

Was done as recommended.

  • Line 37: The (Our) results indicate…

Was done as follows: “The results indicate…”

  • Line 50: Suggestion: replace “…and death” with “…finally leading to death” or similar

Was done as follows:

“The disease is characterized by therapy-resistant diarrhea, progressive emaciation, and a decrease in milk production, finally leading to death…”

  • Line 64-65: Please rephrase, what is meant by “animals”? Species? Individuals? Breeds? Genetic lines?

Line 65: “Individual animals” were meant and included in the sentence.

As Ptb is a multifactorial disease, genetics and other pre-existing conditions are discussed as possible influencing factors for the development of the disease.

In addition, in Line 53 “individual animals” instead of “animals” was included.

  • Line 67: Replace “in the world” by “worldwide”

Was done as recommended (now line 68).

  • Line 121: Delete “from breeder W.”

Was done as recommended.

  • Line 136: “MAP was first detected culturally in farm 1 …”

Was done as follows (now line 143): “…MAP was first detected culturally in herd 1 in January 2018….”

  • Line 147: At what timepoint the environmental samples have been collected?

Was done as follows:

“...Furthermore, environmental samples from various locations in Barn C were sampled in 2020, 2021, and 2022, cultivated and…” (now line 155)

  • Line 115-165: When describing the origin of the purchased goats always add the Ptb status of the herds, if not known please add “unknown Ptb status”. This information is missing for some farms, animals were purchased from.

Was done as recommended and the changes were highlighted in yellow in the text (line 123, 127-128).

  • Line 196: are listed

Was done as recommended (now line 206).

  • Line 204: Clinical symptoms

Was done as recommended (now line 214).

  • Line 264: How were the samples for genotyping selected?

The following text was included into Lines 274 - 279:

“A total of 133 fecal MAP isolates were randomly chosen for genotyping from 121 out of the 186 goats in herd 1 that were detected to be MAP-positive by culture between 2018 and 2022. The MAP isolates from 13 goats with clinical Ptb symptoms selected for genotyping originated from one to four different tissues per goat and included those from the small intestine, various lymph nodes and occasionally from tissues of other organs. All isolates, as well as those from all environmental samples of herd 1…” 

  • Line 438: originated from,

Was done as recommended (now line 454).

  • Line 449: so far only has

Was done as recommended (now line 465).

  • Line 571: The results

Was done as recommended (now line 587).

  • Line 614 and 618: Delete “breeder W.”

Was done. The text was changed as follows:
Line 614, now line 627: “…from goat 84924 (born and purchased in 2011, left the herd 03/2018) …”
Line 618, now line 631:
“…together with other goats from a breeder with unknown Ptb status.”

  • Line 665: There was no evidence for transmission…

Was done (now line 685).

  • Table 1: Write “Mycobacterium avium subsp. paratuberculosis” in the title, or add a key, as it has to be understandable without connection to the text of the manuscript

Instead of “MAP”, “Mycobacterium avium subsp. paratuberculosis” was included in the title of Table 1.

  • Table 2: Include “…tested for Mycobacterium avium subsp. paratuberculosis…” in the title

Was done as recommended.

-----------------------------------------------------------------------------

In addition, some minor changes have been added:
Line 110: Instead of “flocks”, we wrote “herds”

Table S1, line 1013:
Instead “Underlined – goats (n = 12) were sampled twice.” we wrote now: “…goats (n = 12) of which two isolates were genotyped”   

Reviewer 2 Report

Comments and Suggestions for Authors

The authors present a manuscript: "Molecular diversity of Mycobacterium avium subsp. paratuberculosis in dairy goat herds from Thuringia (Germany)". This is an interesting and important work to understand the epidemiology of Mycobacterium avium subsp. paratuberculosis in the region of Thuringia (Germany). Despite only including 4 goat herds, given the scarcity of knowledge in this field, the study makes a relevant scientific contribution.Therefore, the manuscript is subject to publication after minor revisions. Namely

-Change of title to:  Molecular diversity of Mycobacterium avium subsp. paratuberculosis in four dairy goat herds from Thuringia (Germany);

- Line 20 to 23: The study does not allow us to determine the origin of the infection. Thus, the relevance of making assumptions in the Simple Summary is questionable.

-  Line 117 and 158: it is mentioned that"Herd 1 was established as early as 2002 with a mix of goats, sheep and cattle"; and "Herd 3 was established in the early 1990s, first in a former sheep barn". Wouldn't it be important to mention in the discussion the possibility of aprobable link to sheep and not just to cattle? Despite goats may play a more important role than sheep in the transmission and maintenance of MAP.

- line 597 to 599: No bibliographical references are cited. There are recent studies (in the last 5 years) that describe important risk factors in goats such as species (goat), goats belonging to herds with previous wasting disease, accumulation of manure in the herd and absence of perimeter livestock fencing, and others.

- line 665: please change to There was no evidence of transmission of MAP between the goat farms studied.

Comments on the Quality of English Language

Please review the simple summary: line 9 to 13.

Author Response

Thank you for reviewing our manuscript. All changes made in the manuscript have been highlighted in yellow.

Reviewer 2)

Comments and Suggestions for Authors

The authors present a manuscript: "Molecular diversity of Mycobacterium avium subsp. paratuberculosis in dairy goat herds from Thuringia (Germany)". This is an interesting and important work to understand the epidemiology of Mycobacterium avium subsp. paratuberculosis in the region of Thuringia (Germany). Despite only including 4 goat herds, given the scarcity of knowledge in this field, the study makes a relevant scientific contribution. Therefore, the manuscript is subject to publication after minor revisions. Namely

-Change of title to:  Molecular diversity of Mycobacterium avium subsp. paratuberculosis in four dairy goat herds from Thuringia (Germany);

Was done as recommended.

- Line 20 to 23: The study does not allow us to determine the origin of the infection. Thus, the relevance of making assumptions in the Simple Summary is questionable.

Thank you for this important note. From our point of view, it is not possible to trace the true source of the infection, as it may date back 20 - 30 years. We would like to share with the ordinary reader our suspicions about the possible origin of MAP infection.

In answer to the comment of reviewer 1 the sentence has been reworded:

“Introduction of MAP into the herd by uncontrolled animal trade or environmental exposure to MAP in barns that were previously used by cattle with unknown paratuberculosis status are suspected as possible sources of infection.”  

-  Line 117 and 158: it is mentioned that "Herd 1 was established as early as 2002 with a mix of goats, sheep and cattle"; and "Herd 3 was established in the early 1990s, first in a former sheep barn". Wouldn't it be important to mention in the discussion the possibility of a probable link to sheep and not just to cattle? Despite goats may play a more important role than sheep in the transmission and maintenance of MAP.

To consider this important note, the following sentences were included in Lines 661 - 667

“Furthermore, the presence of different ruminant species with unknown Ptb status has been described as another risk factor for MAP infection (Schrott et al., 2023; Iarussi et al., 2019). As described before, herd 1 was established with a mixture of goats, sheep and cattle in 2002, and herd 3 was first established in a former sheep barn in the early 1990s. This also raises the possibility that there might be an epidemiological link to sheep, besides the already discussed one to cattle. Due to the time that has elapsed since the concerned ruminants left the facilities or herds, no further research can be conducted to address this issue.”

In this context in line 649 the following changes were done: Instead: “…and the herd recently also used a former dairy cattle barn… “, we wrote now: “the herd recently used also a former dairy cattle barn…”

- line 597 to 599: No bibliographical references are cited. There are recent studies (in the last 5 years) that describe important risk factors in goats such as species (goat), goats belonging to herds with previous wasting disease, accumulation of manure in the herd and absence of perimeter livestock fencing, and others.

The text section before discussed the problem of why genotype T1 was predominant in herd 1 and whether there are MAP-C genotypes in general that may be more virulent than other genotypes and therefore predominant. The aim of this section was not to discuss all the risk factors for MAP transmission in a goat herd (this is also not the focus of this study). Your comment and question brought this error to our attention and we have corrected it. We have therefore deleted the sentence “Management practices that facilitate the spread of MAP (poor kidding/birthing hygiene, administration of bulk colostrum or milk, deficits in rearing young animals) are more likely to result in the dominance of a single strain on a farm.” The two sections of the text have been merged. What follows is a discussion of the predominance T1 in herd 1 and why this genotype has been able to spread so widely in herd 1.

Based on your comment, in line 662 two additional Reference (Schrott et al., 2023; Iarussi et al., 2019) were included in which different risk factors for infection with MAP are discussed.

To your comment concerning “..describe important risk factors in goats such as species (goat)” we wrote in line 55-56: “There is evidence that goats are more susceptible to MAP than sheep [7], have a shorter clinically unapparent interval, a faster disease progression compared to sheep and cattle [7,8]”.

In addition, we included the following sentences into Introduction:

Line 71-73: “The species (goat) and the absence of perimeter fencing for livestock are considered the main risk for MAP exposure in specific regions [29].”

Line 75-79: “Several management practices (poor kidding/birthing hygiene, administration of bulk colostrum or milk, deficits in rearing young animals, inappropriate ventilation) and co-infections with other pathogens are discussed to facilitate the spread of MAP or MAP infection [26,30,31].”

- line 665: please change to There was no evidence of transmission of MAP between the goat farms studied.

Was done (now line 685).

-Comments on the Quality of English Language

Please review the simple summary: line 9 to 13.

Was done as follows:

“Paratuberculosis (Johne's disease) is a chronic granulomatous enteritis that affects domestic and wild ruminants worldwide, mainly dairy and beef cattle, but also small ruminants such as goats and sheep. To gain a better understanding of the disease's epidemiology and to develop effective control strategies, it is useful to determine the strain diversity of the causative agent, Mycobacterium avium subsp. paratuberculosis (MAP).”

-----------------------------------------------------------------------------

In addition, some minor changes have been added:
Line 110: Instead of “flocks”, we wrote “herds”

Table S1, line 1013:
Instead “Underlined – goats (n = 12) were sampled twice.” we wrote now: “…goats (n = 12) of which two isolates were genotyped”